# A Comparison and Interpretation of Machine Learning Algorithm for the Prediction of Online Purchase Conversion

**Jungwon Lee [1], Okkyung Jung [1], Yunhye Lee [1], Ohsung Kim [1] and Cheol Park [2],***

1   Department of Corporate Management, Korea University, 2511 Sejong-ro, Sejong City 30019, Korea; d2ljw510@naver.com (J.L.); juliewin@korea.ac.kr (O.J.); younhye@korea.ac.kr (Y.L.); osk800@naver.com (O.K.)
2   College of Global Business, Korea University, 2511 Sejong-ro, Sejong City 30019, Korea
*   Correspondence: cpark@korea.ac.kr

**Abstract:** Machine learning technology is recently being applied to various fields. However, in the field of online consumer conversion, research is limited despite the high possibility of machine learning application due to the availability of big data. In this context, we investigate the following three research questions. First, what is the suitable machine learning model for predicting online consumer behavior? Second, what is the good data sampling method for predicting online con-sumer behavior? Third, can we interpret machine learning's online consumer behavior prediction results? We analyze 374,749 online consumer behavior data from Google Merchandise Store, an online shopping mall, and explore research questions. As a result of the empirical analysis, the performance of the ensemble model eXtreme Gradient Boosting model is most suitable for pre-dicting purchase conversion of online consumers, and oversampling is the best method to mitigate data imbalance bias. In addition, by applying explainable artificial intelligence methods to the context of retargeting advertisements, we investigate which consumers are effective in retargeting advertisements. This study theoretically contributes to the marketing and machine learning lit-erature by exploring and answering the problems that arise when applying machine learning models to predicting online consumer conversion. It also contributes to the online advertising literature by exploring consumer characteristics that are effective for retargeting advertisements.

**Keywords:** machine learning; purchase conversion; data imbalance; explainable artificial intelligence

## 1. Introduction

Because prediction of consumer behavior becomes a prerequisite for marketing decision-making, it is considered very important theoretically and practically [1,2]. For example, online shopping mall marketers can improve the marketing performance of re-targeting advertisements by accurately predicting consumers who are likely to purchase among visiting customers [3]. The higher the accuracy of the prediction, the higher the return-on-investment (ROI) of marketing investments can be. Recently, interest in machine learning models has been increasing as a technology that can predict consumer behavior. The Marketing Science Institute (MSI) presented "What is the most effect way to conduct account-based marketing in the face of new online technologies?" as a 2020–2022 Research Priorities Tier 1 research question. Account-based marketing, which recognizes consumers individually and executes marketing, is a topic that needs to be studied urgently to effectively apply new technologies such as machine learning. As such, with the advancement of machine learning technology and increasing interest, researchers in various fields, including business administration, are applying machine learning to prediction problems. Machine learning technology has been studied in various fields such as stock market price prediction [4] and credit rating in the financial sector [5].

One of the most suitable areas to which machine learning for prediction purposes can be applied is in the field of online customer behavior [6]. Marketing literature has understood the consumer's purchasing process step by step through a conversion funnel

model to predict consumer purchasing behavior [7–9]. Unlike the offline environment, the online environment opens up a new opportunity to predict customer behavior through the use of machine learning, as it can identify the consumer journey and various click stream data [10,11]. The customer journey represents a series of stages through which the user gradually goes through the recognition stage as they evaluate alternatives to the actual purchase of the product [12]. Customer journey mapping improves these interactions, resulting in increased sales [13]. Scholars have proposed a variety of frameworks, including logistic regression models [14], game theory-based approaches [15], Bayesian models [16], mutually exciting point process models [17], VAR models [18], and hidden Markov models [19]. However, despite its great potential, our knowledge of predicting online consumer behavior using machine learning is not sufficient. Specifically, the gaps dealt with in this study are as follows.

First, most of the previous studies focused on predicting customer churn using machine learning [6,20,21], but studies that predicted customer's purchase conversion behavior are limited. However, research on consumer behavior in marketing such as a funnel model is ultimately involved in predicting purchase behavior and promoting this process, and in an online environment, the purchase conversion rate is very important to the performance of online shopping malls [22]. Therefore, it is necessary to examine the issues that arise when applying machine learning, which is known to have excellent predictive power, to predicting online consumer behavior.

Second, there is not enough discussion about what machine learning models are suitable for predicting online consumer behavior. All marketing decisions involve predictions of specific outcomes [2]. Therefore, it can be said that the purpose and effect of using machine learning is to improve prediction accuracy [23]. However, prior studies cannot explain which of the various machine learning models is the suitable machine learning model to be used in the context of online consumer behavior. For example, Ballestar et al. [24] used MLP (Multi-Layer Perceptron) ANN (Artificial Neural Network) to predict customer quality targeting E-commerce social networks, but there is a limitation in that it cannot explain which type of machine learning brings excellent results. Exceptionally, Hartmann et al. [25] compared the performance of five machine learning algorithms in the context of text sentiment analysis, but studies conducted in the context of online consumer behavior are insufficient.

Third, comparative analysis was not conducted on the data sampling method suitable for online consumers' purchase conversion. Since most of the marketing data is composed of an unbalanced sample, this bias is an obstacle to the use of machine learning [26]. Therefore, in order to apply machine learning in a marketing context, it is necessary to resolve the imbalance bias. However, most previous studies used only one sampling method, and studies comparing two or more methods are insufficient.

Fourth, there is insufficient discussion on how to interpret the results of machine learning models in the context of marketing. Machine learning models have an excellent predictive performance, but have a limitation in that they cannot explain the relationship between predictors and results [1]. The problem of interpretability is a very important issue in the practical use of marketing, as it can reduce user confidence in machine learning and negatively affect the use of machine learning prediction results. Explainable Artificial Intelligence (XAI) is being developed to solve this problem [27], but studies discussing the applicability of the actual marketing context are insufficient.

This study utilized log data from the Google Merchandise Store to explore these research questions. Through data structuring, a total of 374,749 customer decision journey data and 687 explanatory variables were measured and analyzed with a machine learning model. The detailed analysis process is as follows. First, the performance of eight major machine learning models was compared and analyzed using the Caret package. Second, as a method of mitigating data imbalance, (1) internal algorithm, (2) under-sampling (3), and over-sampling were performed, and the predictive performance of the machine learning model was compared and analyzed. Third, XAI was applied to increase the

possibility of interpreting machine learning models. Specifically, we intended to train a purchase prediction model in the context of retargeting advertisements and explore new knowledge through XAI. Retargeting advertisements are online advertisements targeting consumers based on previous Internet activity in situations where online advertisements do not lead to sales or conversions. These advertisements are widely used in practice, but we do not have enough knowledge regarding them [3].

The results of this study have the following theoretical and practical implications. First, unlike previous studies that explored a small number of machine learning models, this study contributed to the machine learning methodology of online marketing by comparing 8 machine learning model algorithms. Second, there is a theoretical and practical contribution by comparing the sampling method, which has obtained relatively little interest from machine learning researchers in the online marketing context. Since online consumer purchase data inevitably involves imbalance, the results of this study can serve as an initial guideline for future research and use of machine learning by practitioners. Third, and most importantly, it explored the application of XAI. We analyzed which consumers are effective for retargeting advertisements through various XAI methodologies (e.g., Shapley Additive exPlanations). Moreover, prediction results in the context of individual consumers can be interpreted using eXtreme Gradient Boosting (XGB) Explainer. The application of XAI provides an opportunity to expand knowledge through machine learning research by linking machine learning with the online consumer behavior literature.

The remainder of this study is organized as follows: First, we review prior studies and limitations related to this study on funnel model and online conversion behavior, which are important theories for understanding consumer behavior. In addition, it derives research questions by reviewing machine learning research in the context of marketing. Second, the data collection method and characteristics of the data are discussed. Third, it proposes a research method suitable for solving the research questions and discusses the analysis results. Fourth, the theoretical and practical implications of this study are discussed, and future research directions are presented.

## 2. Theoretical Background

### 2.1. Funnel Model and Online Purchase Conversion

In the marketing literature, the conversion funnel model has been studied as an important theory for understanding consumer decision-making and behavior [7–9], and is used as a core framework for marketing decision-making [3]. Researchers have tried to understand consumer behavior through various transformations in the basic funnel model structure such as attention, interest, decision, and purchase [14]. In recent years, online data has been used to more easily identify the funnel stage of online consumers compared to offline [3,28]. Although it is very difficult to analyze what stage a consumer is at in an offline environment, Internet-based click stream data contains information of various consumers, so the funnel stage can be analyzed more accurately.

Researchers analyzed the relationship between consumer engagement behavior and conversion such as duration time [29], page view [30], and search depth [31]. These engagement variables reflect consumer interest while visiting a website [32], and have been studied as an important quality factor of consumer information processing in search and purchase situations [33]. A number of researchers have reported that there is a positive relationship between engagement behavior and consumer conversion [34]. In addition, the relationship between the movement pattern within the website [35] or consumer visit history [31] and conversion was analyzed.

Recently, in a new context such as social media, the measurement of more in-depth variables such as mouse scrolls has been applied to analysis. Goldstein et al. [36] measured search diversity, which means how many different pages a customer searches, and reported that as the search diversity increases, the likelihood of purchase decreases. Lo et al. [37] analyzed the purchase possibility of customers in Pinterest by dividing them into short-term and long-term purchase intentions. According to the analysis results, users with

long-term purchase intentions clicked and saved more external contents, and customers with short-term purchase intentions tended to perform relatively more search actions. In addition, Guo and Agichtein [38] analyzed the relationship between customer purchase intentions and mouse-related data. They found that customers who are willing to buy scroll more.

The previous studies are summarized as follows. Most previous studies are based on a funnel model and analyzed the relationship between customer engagement metrics and conversions using clickstream data. In predicting online conversion behavior, the researcher predicts page views, duration time [39], the sequence of various actions involving Search diversity [36], and scrolling behavior [30]. However, prediction of online consumer behavior using machine learning models has not been sufficiently studied for its potential [38]. Most studies using machine learning focus on predicting customer churn [20,21], and studies predicting customer conversion behavior are limited. Therefore, in this study, we review previous research on predictive machine learning models in a marketing context and present important research questions.

### 2.2. Machine Learning Models

The application of machine learning in a marketing context is emerging as an important trend due to the availability of big data along with a complex marketing environment that becomes increasingly difficult to predict. Samuel [40] defined machine learning as "field of study that gives computers the ability to learn without being explicitly programmed". As such, machine learning is composed of mechanisms that reflect specific work experiences to improve its performance and evaluate it. In the marketing literature, studies using machine learning have been reported (for a review, see Ma and Sun [41]). Previous studies include prediction [1,42], feature extraction [43], technical interpretation [44], Prescriptive analysis [45], and optimization [46]. In addition, in terms of machine learning research methods, SVM (Support-vector machine; Cui and Curry [1]), Topic modeling [44], Ensemble trees [47], deep learning [24], and other machine learning models were also studied. The research results used are reported. The previous studies on these major machine learning models are as follows.

First, one of the first machine learning models introduced in marketing is SVM. In the context of marketing, Cui and Curry [1] compared SVM with the multinomial logit model and presented the result that SVM had better predictive performance. Specifically, the multinomial logit model is more suitable for presenting implications, but they found that SVM is more suitable for environments dealing with large-scale data. These results point to a limitation in that the machine learning model is very good in predictive performance, but cannot explain the relationship between the explanatory variable and the predicted result.

Second, the deep learning model is a machine learning model most widely used in recent marketing research, and is used for text and image data analysis. For example, Liu et al. [48] analyzed consumer reviews and reported that aesthetics and price influence conversion. In addition, Chakraborty et al. [49] developed a Hybrid CNN-LSTM model to extract emotional characteristics from text data, and showed that it solves difficult emotion classification problems well for Yelp reviews. Zhang et al. [50] analyzed the effect of images on Airbnb's accommodation demand by using deep learning. The authors classified the quality of images using a convolution neural network (CNN), and found that high-quality photos increase the demand for the accommodation. Similarly, Zhang and Luo [51] analyzed the photos of Yelp reviews and presented the results that photos could more accurately predict the survival of restaurants than review content.

Third, the Ensemble method is an algorithm that combines several individual learners and is characterized by high prediction accuracy. Stacking, bagging, and boosting are used as general methods. Stacking improves accuracy by using a linear combination of individual explanatory variables [47], while in bagging, each individual learning tree is obtained using bootstrap samples, and the predictions of individual learning trees are aggregated to produce the final prediction result. On the other hand, in boosting,

individual learning trees are trained sequentially and a stronger learning tree is created according to each accuracy. Also, adaptive boosting is a method in which subsequent learning is adjusted for previously misclassified content. In sum, in the case of bagging, it learns in parallel, and in the case of boosting, it learns sequentially, and weights are given to the result after learning is completed. In general, boosting results in high performance for individual decision trees, but it is slow and there is a high probability of overfitting that over-learns the training data.

Popular ensemble models include Random-Forest [52] and Gradient-Boosted Tree (GBM; [53]). Each uses bagging and boosting models. In the random forest, individual trees are constructed from bootstrap samples of the original data, and each divided tree is randomly assigned input variables to reduce correlation. Finally, the prediction results of individual trees are averaged to calculate the final prediction. In GBM, several trees are trained sequentially, and each tree improves accuracy by reducing errors in previously applied trees. Recognizing scarcity, XGB is a model that has won numerous data science competitions in Kaggle [54].

Prior research on machine learning for predictive purposes reports that marketing decision-making can be supported using various machine learning methodologies. However, studies related to online consumer conversion prediction are relatively limited, and insufficient research has been conducted on which machine learning methodologies are effective in the context of online consumer behavior prediction (Table 1). In addition, most of the marketing prediction problems are accompanied by class imbalance, which reduces predictive performance. Therefore, for effective use of machine learning, comparative analysis of methodologies that can mitigate imbalances is needed. Finally, as pointed out by the study of Cui and Curry [1], machine learning models have difficulty in interpreting the relationship between explanatory variables and prediction results. In particular, research using XAI is needed for retargeting advertisements that have high utilization of machine learning but have not been sufficiently studied. Through this analysis, it will be possible to connect the machine learning literature with the marketing literature and provide new knowledge for retargeting advertisements.

**Table 1.** Comparison between related research and this research.

| No | Researcher(s) | Model | Sampling | XAI | Main Result |
|---|---|---|---|---|---|
| 1 | Cui and Curry [1] | SVM | - | - | Verifying that machine learning-based SVM outperforms traditional prediction models in various marketing prediction environments |
| 2 | Huang and Luo [30] | Fuzzy SVM | - | - | Proposed a framework that can analyze consumer preferences for products with complex characteristics through Fuzzy SVM |
| 3 | Jacobs et al. [55] | LDA, MDM | - | - | Using LDA (Latent Dirichlet Allocation) and MDM (Mixtures of Dirichlet-Multinomials), propose a methodology to predict which products customers will purchase |
| 4 | Miguéis et al. [26] | Random Forest | 3 methods | - | A comparison of the performance of the sampling method was conducted for the prediction of consumer response to direct marketing for the banking industry |
| 5 | Ballestar et al. [24] | MLP ANN | - | - | Using Multilayer Perceptron Artificial Neural Network to predict customer quality for e-commerce social networks |
| 6 | Hartmann et al. [25] | 5 models | - | - | Compare 5 machine learning models as a methodology for classifying the sentiment of text (SVM, Random forest, Naive Bayes, ANN, and KNN) |
| 7 | This study | 8 models | 3 methods | 3 methods | Comparing suitable machine learning algorithms and sampling methods in the context of online consumer behavior, and applying XAI to explore effective consumer characteristics for retargeting advertisements |

*2.3. Research Questions*

Previous studies analyzing predictive machine learning model algorithms in the marketing context are shown in Table 1 above. Most of the previous studies have only adopted and analyzed one machine learning model, but studies that have performed comparisons between machine learning models are limited. In addition, there are insufficient studies to verify the performance of the sampling method, a method capable of dealing with data imbalance. Finally, there are not enough studies applying XAI. Therefore, in this study, we propose a research problem focusing on these three limitations.

2.3.1. Performance of Predictive Machine Learning Algorithms

Politz and Deming [2] mentioned the importance of predictive models in marketing half a century ago. Prediction accuracy in marketing is very important because it directly affects marketing performance [23]. However, prior machine learning studies have not been sufficiently studied in terms of online consumer behavior prediction. In other words, current knowledge cannot tell which machine learning model is suited for predicting online conversions. For example, Ballestar et al. [24] used MLP ANN to predict customer quality targeting e-commerce social networks, but there is a limitation in that it cannot explain which form of machine learning provides excellent results. Therefore, the following research questions was considered for this study:

Research Question 1: What machine learning models are suitable for predicting online consumer behavior?

2.3.2. Data Imbalance Problem

Since the machine learning classification algorithm assumes that data are evenly distributed among different classes, predictive performance decreases in the case of unbalanced data [56]. However, since most of the marketing data are composed of an unbalanced sample, this has a negative impact on machine learning performance [26]. In the case of online consumer data collected in this study, the number of converting customers is only about 2.29% of the total. Previous studies have suggested internal algorithms and external approaches to data as a way to mitigate the negative effects of data imbalance on prediction accuracy [56,57]. The former is to set weights to mitigate imbalances in machine learning algorithms. In other words, a weight is given to prediction of a small number of class groups (i.e., conversion classes). The latter is to use a sampling method before putting the data into training. Data sampling methods include over-sampling and down-sampling.

Under-sampling is a method of balancing the original number of minority classes by removing samples from the majority class. Previous researchers tried to develop an effective method by suggesting various under-sampling methods. For example, Yen and Lee [58] proposed a cluster-based under-sampling approach to improve the classification accuracy of minority classes. In addition, Liu et al. [58] proposed an under-sampling method based on two pieces of methods: Easy Ensemble and Balance Cascade.

On the other hand, the over-sampling method is a method of replicating samples of a minority class or creating new samples to balance them. However, it has been reported that minority-class sampling is relatively inefficient [26]. On the other hand, SMOTE (the Synthetic Minority Over-sampling Technique) introduced by Chawla et al. [57] is considered a relatively effective over-sampling technique as a method of balancing by randomly inserting the nearest neighbor pair in a minority class. According to a study of Miguéis et al. [26], it is possible to predict the response to direct marketing, and they found that the predictive performance can be improved by using the SMOTE sampling method. However, in the context of online consumer conversion, the discussion on the sampling method was not sufficiently conducted [59]. Therefore, the following research question was considered:

Research Question 2: What is a good data sampling method for predicting online consumer behavior?

2.3.3. Explainable Machine Learning

As suggested by the work of Cui and Curry [1], machine learning models have difficulty in interpreting the relationship between predictors and outcomes. Algorithms of machine learning have complex structures and learning mechanisms, making it difficult for humans to interpret their relationships [27]. This problem of interpretability negatively affects the user's confidence in machine learning, and it is an obstacle to the use of machine learning prediction results. Because of this problem, machine learning researchers are suggesting interpretable machine learning techniques [27]. However, up to now, there have not been enough empirical studies applying XAI in the context of marketing. Therefore, the following research problem was considered:

Research Question 3: Can we interpret machine learning's online consumer behavior prediction results?

## 3. Data

### 3.1. Data Collection

The data was collected from Google Analytics data from the Google Merchandise Store. The data collection period is from 1 August 2016 to 15 October 2018, and the data was published as contest data on Kaggle.com. The Google Merchandise Store is a site that sells Google's souvenirs, and because consumers from all over the world visit and purchase it, it is suitable for obtaining results that have sufficient data and can be generalized to the entire world [60]. In this study, basic information was collected by accessing Google Analytics, and data provided in the Kaggle contest were used for individual visitor information. These data were incorporated. The Table 2 below summarizes the basic information of the data. The total number of users is 1,668,539, and the funnel was divided into organic search, social media, direct visit, referrer, search advertisement, display advertisement, affiliate, and others.

**Table 2.** Collected data.

| Website Profile | | Users by Channel | | Performance | |
|---|---|---|---|---|---|
| Period | 2016.8~2018.4 | Organic search | 759,171 | Revenue | USD 9,203,274.38 |
| User | 1,668,539 | Social media | 363,521 | Conversion rate | 2.29% |
| New User | 1,648,685 | Direct | 281,996 | Transaction | 51,498 |
| Session | 2,250,073 | Referrer | 184,878 | Average purchase amount | USD 178.71 |
| Session per user | 1.35 | Search ad | 44,078 | | |
| Page view per session | 4.22 | Display ad | 40,635 | | |
| Bounce rate | 47.23 | Affiliate | 35,926 | | |
| Average session time | 2 min and 27 s | Other | 105 | | |
| Mobile ratio | 27.59% | | | | |

### 3.2. Sample Characteristics

Next, the characteristics of the sample were analyzed. Specifically, data were classified into purchasing customers and non-purchasing customers, and the averages were compared to see if there were any differences in the explanatory variables constituting the session. As a result of the analysis, it was found that most of the variables were different between the non-conversion session and the conversion session (Table 3). In particular, there were significant differences in visit quality indicators such as the inflow channel, page view, and duration. Therefore, it was determined that it is reasonable to analyze the relationship between the explanatory variable and the conversion by using the data. These results are summarized in Table 3 below.

**Table 3.** Comparison of consumer session data with or without conversion.

| Variables | Non-Purchasing Customer | Purchasing Customer | F |
|---|---|---|---|
| New visit | 0.7694 | 0.3930 | 14,600 ** |
| Number of visits | 2.3165 | 4.0298 | 616.1 ** |
| Pageview | 3.4407 | 26.9106 | 281,045 ** |
| Duration time | 113.9927 | 1022.44 | 129,741 ** |
| Organic search | 0.4340 | 0.2973 | 1398 ** |
| Social media | 0.2099 | 0.0086 | 4531 ** |
| Direct | 0.1598 | 0.1802 | 58.17 ** |
| Referrer | 0.1199 | 0.4632 | 20,192 ** |
| Search ad | 0.0265 | 0.0383 | 98.06 ** |
| Display ad | 0.0302 | 0.0110 | 230.7 ** |
| Affiliate | 0.0194 | 0.0009 | 334.5 |
| Other | $8.048395 \times 10^{-5}$ | $5.388221 \times 10^{-5}$ | 0.162 |

\* $p < 0.05$, \*\* $p < 0.01$.

### 3.3. Measurement

In order to consider and analyze the consumer's entire purchasing process, we needed to reconstruct the current data composed of individual visits into customer journey data [60]. That is, if consumer A accesses the site three times, evaluates the product, and purchases the product on the fourth visit, the data on the fourth visit must contain information about the previous visit (e.g., number of visits, number of previous conversions). Since Google Analytics tracks the consumer's unique number for each session data, the data is linked based on this unique number. Through this process, the final consumer journey data result was 374,749. The inflow channel measured the last access channel. Variables composed of a continuous scale measured the number of visits, page views, session quality, duration time, and number of previous conversions. The dependent variable is a binary variable, which means that the interpretation respects a binary discrete choice modelling approach (1 if purchase occurs; 0 if purchase does not occur). In addition, as categorical variables, browser types, inflow channel, operating system types, device types, access countries, and access cities were measured. Categorical variables were measured by converting them into dummy variables. Finally, 687 variables were measured including dummy variables. These results are summarized in the Table 4 below.

**Table 4.** Consumer journey data.

| Variables | Min | Max | Mean | Standard Deviation |
|---|---|---|---|---|
| Purchase | 0 | 1 | 0.05 | 0.31 |
| Number of visits | 1 | 456 | 5.71 | 18.93 |
| Page view | 0 | 500 | 5.45 | 9.30 |
| Hits | 1 | 500 | 6.80 | 13.06 |
| Session quality | 0 | 100 | 3.59 | 12.59 |
| Duration time | 0 | 19,017 | 223.50 | 520.13 |
| Number of previous conversions | 0 | 21 | 0.02 | 0.16 |
| Inflow channel | 8 dummy variables (e.g., Organic search, Social media, Direct, Referral, Search ad) | | | |
| Browser | 33 dummy variables (e.g., Chrome, Safari, Edge, Samsung Internet, Opera Mini) | | | |
| Operating system | 19 dummy variables (e.g., Macintosh, iOS, Chrome OS, Windows, Tizen, Samsung, Xbox) | | | |

**Table 4.** *Cont.*

| Variables | Min | Max | Mean | Standard Deviation |
|---|---|---|---|---|
| Device | 3 dummy variables (e.g., Desktop, Mobile, Tablet) | | | |
| Country | 196 dummy variables (e.g., United States, Canada, Philippines, Mexico, India, Japan) | | | |
| City | 421 dummy variables (e.g., New York, California, Massachusetts, Gujarat, Taipei City) | | | |
| Grand total | Variables: 687, number of data: 374,749 | | | |

## 4. Result

### 4.1. Research Question 1

The aim of Research question 1 was to analyze which algorithm is suitable for predicting online consumer behavior. In this study, conversions were measured and analyzed as the dependent variable. There is a total of 8 major machine learning algorithms, and the major algorithm models discussed in the theoretical background were selected. Specifically, Classification Tree [61], Artificial Neural Network [62], KNN (K-Nearest Neighbor algorithm; [63]), logistic regression analysis, SVM [1], random forest [52], GBM [53], and XGB [54] methods were selected. As an analysis method, machine learning algorithms were compared using R's CARET package. This package contains functions that can simplify the process of model development and evaluation, so it is effective for comparing model performance (Figure 1). The learning function uses Re-sampling to evaluate the effect of the model's adjustment parameters on the performance, selects the optimal model from these parameters, and estimates the model performance from the training set. In this study, 5 candidate values for the tuning parameter were set and K-fold cross validation of the data was set to 5.

```
1  Define sets of model parameter values to evaluate
2  for each parameter set do
3      for each resampling iteration do
4          Hold–out specific samples
5          [Optional] Pre–process the data
6          Fit the model on the remainder
7          Predict the hold–out samples
8      end
9      Calculate the average performance across hold–out predictions
10 end
11 Determine the optimal parameter set
12 Fit the final model to all the training data using the optimal parameter set
```

**Figure 1.** CARET package algorithm.

In addition, AUC (Area Under the Curve; The area under a receiver operating characteristic [ROC] curve, is a single scalar value that measures the overall performance of a binary classifier) was selected and performed as an index of machine learning performance comparison [64]. In this study, sensitivity, specificity, positive predictive value (PPV), negative predictive value (NPV), and prevalence were used to compare the performance of machine learning with reference to the study of Vermeer et al. [42]. These indicators are based on the concept of measurement as to whether predictions are classified incorrectly or correctly [65]. The measurement concept is 1) TP (True Positive) when the actual positive (conversion) is determined as positive, 2) FP (False Positive) when the actual negative (non-conversion) is determined as positive, and 3) negative. The latter is divided into FN (False Negative) and 4) True Negative (TN), that is positive as negative. In addition, using this measurement concept, the model was evaluated by calculating the index as follows.

- Sensitivity: $\frac{TP}{TP+FN} \times 100$
- Specificity: $\frac{TN}{FN+TN} \times 100$
- Accuracy: $\frac{TP+TN}{TP+FP+FN+TN} \times 100$
- Pose Pred Value (PPV): $\frac{TP}{TP+FN} \times 100$
- Neg Pred Value (NPV): $\frac{TN}{FN+TN} \times 100$
- Prevalence: $\frac{TP+FN}{TP+FP+FN+TN} \times 100$

The analysis results are shown in the table below. For the sensitivity, LOGIT analysis was the most effective, and NNET showed the best performance for the specificity, but in the case of ROC (Receiver operating characteristic), which is the overall performance, XGB showed the best performance (Table 5; Figure 2). Therefore, the best machine learning algorithm for learning and predicting online consumer behavior data was analyzed with XGB.

**Table 5.** Machine learning algorithm performance comparison.

| No | Machine Learning Model | AUC | Sensitivity | Specificity |
|:--:|:--:|:--:|:--:|:--:|
| 1 | Classification tree (TREE) | 0.8133 | 0.7030 | 0.8419 |
| 2 | Artificial neural network (NNET) | 0.8367 | 0.6744 | 0.8815 |
| 3 | K-Nearest-Neighbor (KNN) | 0.8323 | 0.6913 | 0.8332 |
| 4 | Logistic Regression (LOGIT) | 0.7386 | 0.7410 | 0.7113 |
| 5 | Support vector machine with linear kernel (SVML) | 0.8318 | 0.6881 | 0.8542 |
| 6 | Random forest (RF) | 0.8544 | 0.7017 | 0.8702 |
| 7 | Gradient Boosting Algorithm (GBM) | 0.8640 | 0.7356 | 0.8451 |
| 8 | eXtreme Gradient Boosting (XGB) | 0.8643 | 0.7235 | 0.8560 |

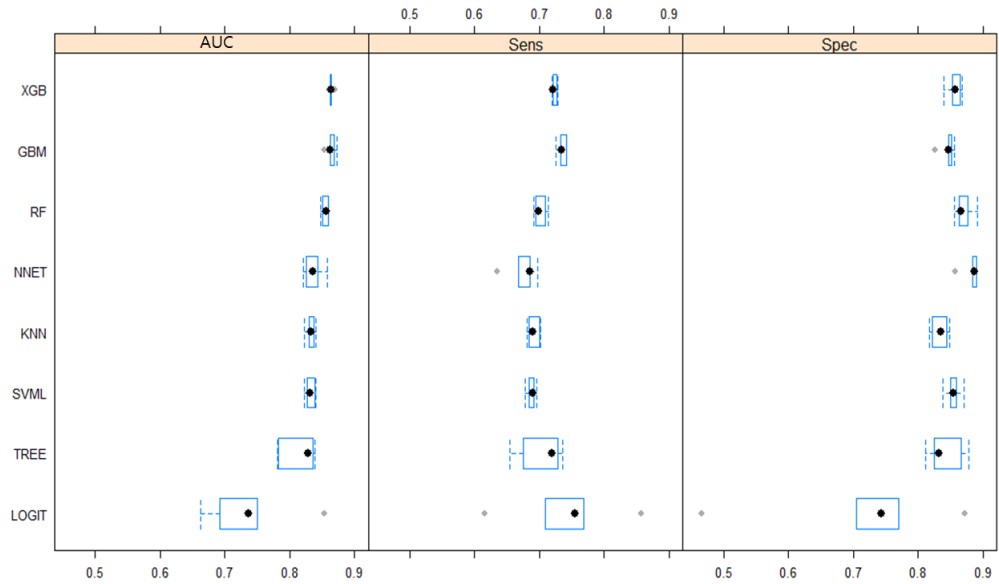

**Figure 2.** Comparison of machine learning model performance.

A visualization of these results is shown in the graph below. In the case of XGB, it confirms that the AUC performance is the best compared to other models. Therefore, subsequent research question analysis was performed through XGB. The objective function of XGB is as follows [54]. The regularized objective function contains two parts: the training loss function $\iota$ and the regularization term $\Omega$. The training loss $l$ measures the difference between the predicted value $\hat{y}_i$ and the true value $y_i$. The regularization term $\Omega$

measures the complexity of the model, which helps to smooth the final learnt weight to avoid overfitting.

$$obj(\theta) = \sum_i^n l(y_i, \hat{y}_i) + \sum_i^k \Omega(f_k) \tag{1}$$

$$Gain = \frac{1}{2}\left[\frac{G_L^2}{H_L + \lambda} + \frac{G_R^2}{H_R + \lambda} - \frac{(G_L + R)^2}{H_L + H_R + \lambda}\right] - \gamma \tag{2}$$

Beginning with a tree with a depth of 0, the tree continues to grow as more information is acquired when pruning (Greedy Learning of the Tree). The Gain function Equation (2) consists of the left child node function, the right child node function, and the score when undivided.

*4.2. Research Question 2*

Addressing Research problem 2 involved analyzing the appropriate sampling method for learning and predicting online consumer behavior by comparing predictive performance according to the sampling method. In this study, under-sampling and over-sampling were performed based on previous studies, and the SMOTE methodology was adopted for the over-sampling method by referring to previous studies [18]. In addition, the machine learning algorithm was implemented with XGB with reference to the analysis result of Research question 1. First, the original data was divided into training data and verification data at a ratio of 7:3. The predictive performance was analyzed by sequentially applying each sampling method to the following training data. The change in the number of data according to the sampling method is shown in the Table 6 below. The original data shows an imbalance with a very small amount of data in the conversion case, and it could be confirmed that it is balanced 1:1 after under-sampling. Next, over-sampling adjusted the imbalance by adding the number of conversion data and removing some non-conversion data. In addition to the sampling method, there is a method of setting weights in the learning algorithm to alleviate the data imbalance. In this study, in order to compare the internal algorithm method and the sampling method, a method of assigning weights according to the ratio of conversion data in the original data analysis was applied.

**Table 6.** Training data according to sampling method.

| No | Sampling Method | Number of Data | | |
|:---:|:---:|:---:|:---:|:---:|
| | | Total | Purchased | Un-Purchased |
| 1 | Original data | 262,324 | 6303 | 256,021 |
| 2 | Under-sampling | 12,606 | 6303 | 6303 |
| 3 | Over-sampling (SMOTE) | 264,726 | 132,363 | 132,363 |

On the other hand, the performance of machine learning varies according to the setting of learning parameters. Therefore, after applying each sampling method, it is necessary to reset the optimal parameters for the data. Therefore, in this study, after applying each sampling method, we used the Bayesian Optimization Package (rBayesianOptimization). The best parameters for each data were searched for six times and the parameters with the highest performance were applied. The results of the parameter search are included in Appendix A.

Model training was performed on each training data, and the predictive performance of the model was evaluated through the verification data. The performance evaluation was verified by an OOB (Out-of-bag; [66]) method that verifies through data not included in the training. As a result of the analysis, it was found that the performance of the over-sampling (i.e., SMOTE) was the best (Table 7). On the other hand, under-sampling was found to have a lower performance than the original data to which the internal algorithm method was applied.

**Table 7.** Comparison of XGB performance according to sampling method.

|  | Original Data | Under-Sampling | Over-Sampling (SMOTE) |
|---|---|---|---|
| Accuracy (95% CI) | 0.7348 (0.7322, 0.7374) | 0.7324 (0.7298, 0.7350) | 0.7417 (0.7391, 0.7442) |
| Sensitivity | 0.7320 | 0.7296 | 0.7392 |
| Specificity | 0.8501 | 0.8444 | 0.8430 |
| PPV | 0.9950 | 0.9948 | 0.9949 |
| NPV | 0.0714 | 0.0704 | 0.0727 |
| Prevalence | 0.9763 | 0.9763 | 0.9763 |

The ROC graph of XGB classification performance using over-sampling data is as follows (Figure 3). The AUC was 79.11%, which was judged to indicate a good classification performance [64].

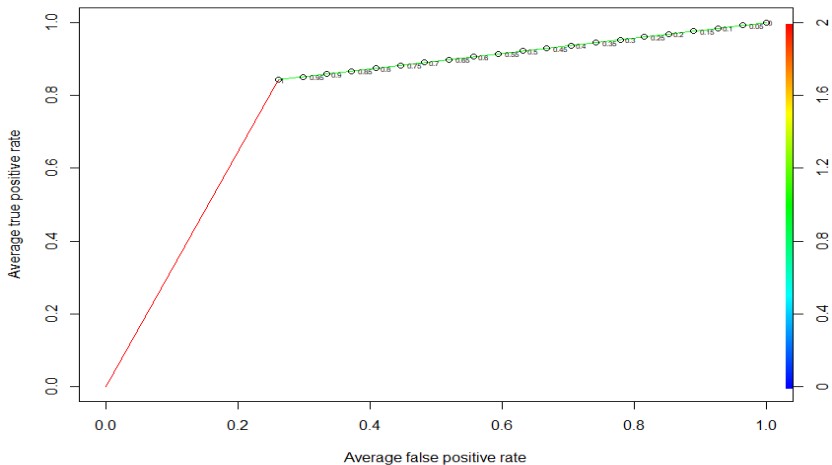

**Figure 3.** ROC Curve.

### 4.3. Research Question 3

Addressing Research question 3 involved verifying whether machine learning can interpret online consumer behavior prediction results. Recently, machine learning researchers have been proposing XAI for this purpose [27]. The overall purpose of this study was to identify problems arising in the process of analyzing online consumer behavior data through machine learning models, to apply them to marketing decision making, and to present the most appropriate methodology. This chapter focuses on the context of retargeting advertisements to explore the applicability of XAI. Retargeting advertisements are personalized advertisements and have a deep relationship with machine learning predictions, but sufficient research has not been presented [3]. Specifically, we predicted the likelihood of conversion when consumers return through retargeting ads. After dividing all 18,340 data into 7:3, we trained 23,885 by applying SMOTE sampling to 12,838 train data sets. There are 5502 test data sets. Analysis using XGB and SMOTE sampling showed 71% accuracy in OOB verification (Table 8). By interpreting it with various XAI methodologies, we intend to present the usefulness of XAI and explore new knowledge about retargeting advertising.

**Table 8.** XGB model performance in the context of retargeting ads.

| Criteria | Performance |
|---|---|
| Accuracy | 0.7147 |
| (95% CI) | (0.7120, 0.7173) |
| Sensitivity | 0.7143 |
| Specificity | 0.7276 |
| PPV | 0.9908 |
| NPV | 0.0581 |
| Prevalence | 0.9763 |
| Detection Rate | 0.6974 |
| Detection Prevalence | 0.7038 |
| Balanced Accuracy | 0.7209 |

For this chapter, two aspects of XAI were applied. First, from a global perspective, XGB Importance analysis and SHAP (Shapley Additive exPlanations) were used. XGB Importance analysis identifies the importance of variables in training the model. The algorithm counts out the importance by employing "gain", "frequency", and "cover". Gain is the main reference factor of the importance of a feature in the tree branches. Frequency, which is a simple version of gain, is the number of a feature in all constructed trees. Cover is the relative value of a feature observation.

SHAP, proposed by Lundberg and Lee [67], was used to interpret the output of the model. SHAP is based on game theory [68] and local explanations [69], and it offers a means to estimate the contribution of each feature. To use this method, assume an XGB model where a group $N$ (with n features) is used to predict an output ($N$). In SHAP, the contribution of each feature ($\phi_i$ is contribution of feature $i$) on the model output $v(N)$ is allocated based on their marginal contribution [70]. More details of the tree SHAP algorithm can be referred to [67]. Based on several axioms to help fairly allocate the contribution of each feature, shapely values are determined through:

$$\phi_i = \sum_{S \subseteq N\{i\}} \frac{|S|!(n - |S| - 1)!}{n!} [v(S \cup \{i\}) - v(S)] \tag{3}$$

As Tree SHAP values are derived from an individualized model interpretation approach, an individualized interpretation for each sample can be obtained from the model [71]. Figure 4 presents some insights into how the contribution of an individual feature on the model output is affected by its value. Each point in the SHAP Importance Analysis plot is the Shapley value and observation value for the characteristic, the x-axis is determined by the Shapley value, and the y-axis is determined by the characteristic. The color represents the value of the characteristic from low to high, and the variables are sorted according to importance. As a result of the analysis, XGB Importance analysis and SHAP analysis were found to be similar (Figure 4). Duration time, device type, number of hits, number of visits, session quality, page views, and inflow channels were analyzed as important factors in that order.

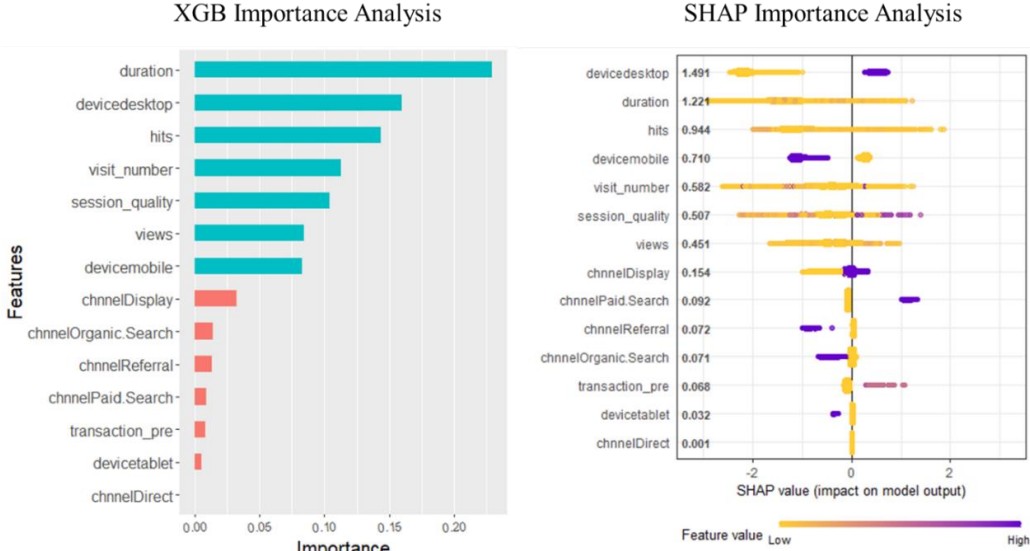

**Figure 4.** Analysis of the importance of variables.

The results of the SHAP importance analysis show the direction of the variable's impact on the likelihood of conversion. The variables that increase the likelihood of purchasing through retargeting ads were connected devices (PC), session quality, display ads, search ads, and past purchasing experiences. On the other hand, it was found that device (mobile), referrer channel, and organic search negatively affected the possibility of purchasing through retargeting advertisements.

However, there are variables that make it difficult to clarify the relationship (e.g., duration). In this case, the SHAP feature dependence graph can show the change of the SHAP value according to the change of the variable value. SHAP feature dependence displays the Shapley value corresponding to the x-axis as the variable value for the observed value on the y-axis.

Figure 5 shows the SHAP value according to the value of each variable. The variables of visit number, session quality, duration, and page views show a nonlinear relationship with the SHAP value. This means that the consumer who gave up the purchase in the previous session must have an appropriate level of prior knowledge and interest in order to lead to purchases through retargeting advertisements, and the retargeting effect does not increase in proportion to the interaction between the consumer and the shopping mall. For example, the page view appears in an inverse relationship with the SHAP value, which means that in order to increase the effectiveness of the retargeting advertisement, an appropriate level should be selected rather than obtaining too few or too many page views.

Second, to obtain the perspective of individual consumers, XGB Explainer was used. This package was created by David Foster in August 2017, and allows for analyzing the weight of each explanatory variable in the final forecast result. The XGB model can interpret individual predicted results by applying the weight of individual variables to the predicted results as log-odds to Equation (4) below. This represents the influence of each variable on all decision trees in the ensemble process as log-odds, and shows the process of deriving the final probability value by summing them through a waterfall chart for each data.

$$\ln\left(odds = \frac{p}{1-p}\right) \tag{4}$$

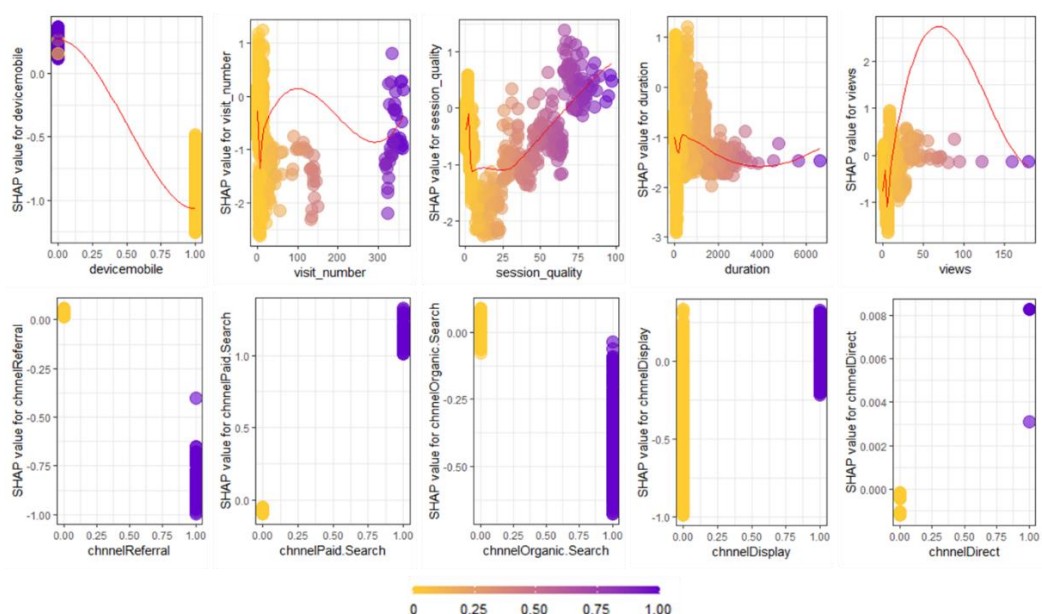

**Figure 5.** SHAP dependence plot.

Combining the effects of each of these explanatory variables, the consumer's likelihood of purchase was calculated at 92.99%, and the consumer actually purchased the product from a visit through retargeting advertising. By applying these interpretable machine learning techniques, the reasons for the machine learning prediction results are interpreted as the influence of individual explanatory variables so that practitioners can understand the machine learning model and get meaningful implications. For instance, the coefficient associated with duration = 705 in Figure 6 is equal to $-0.28$. Assuming that 705 is a measure whose unit is in seconds, this means that, in comparison to the base alternative (that is, in comparison to the decision of not purchasing), an increase in the independent variable (that is, an increase in the duration of the purchase from 705 s to 706 s) makes the individual's decision of purchasing less likely. Hence, the interpretation of this coefficient suggests that extra time spent by an individual on purchasing decreases the likelihood of making a purchase.

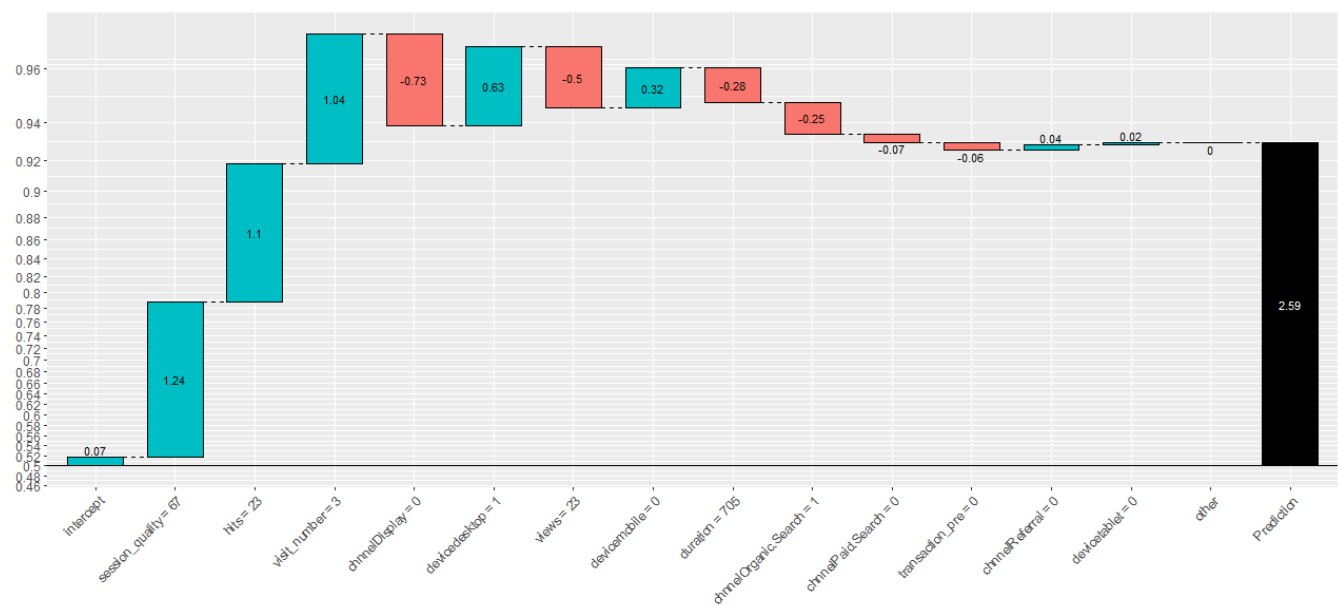

**Figure 6.** Application of interpretable machine learning.

## 5. Conclusions and Implications

*5.1. Summary of Research Results*

The results of this study are as follows. First, the XGB model is the most suitable machine learning model for predicting online consumers' purchase conversion. According to previous research, the ensemble method is a meta-learning algorithm that combines several individual learners and is known to have superior predictive performance compared to other algorithms [72].

In the context of online consumer conversion, this study shows that the predictive power of the ensemble model is superior to other algorithms, and the XGB model that uses a boosting method and recognizes scarcity is the best.

Second, the oversampling method is most suitable for the data imbalance problem that occurs in the context of online consumer behavior. SMOTE, which is the over-sampling method adopted in this study, can be interpreted as having higher performance than the random sampling method by randomly inserting the nearest neighbor pair in the minority class to balance it [26].

Third, it is commonly expected that the possibility of explaining machine learning results in a marketing context can be enhanced by using explainable machine learning technologies. Algorithms of machine learning have complex structures and learning mechanisms, making it difficult for humans to interpret their relationships [27]. In this study, XAI was applied from two perspectives. Specifically, we analyze the importance of each predictor from a global perspective through XGB im-portance analysis and SHAP analysis. In addition, by applying XGB Explainer from the perspective of individual consumers, it was found to be possible to analyze the prediction results of individual consumers by decomposing the contribution of each predictor. In addition, by applying XAI in the context of retargeting advertisements, we found that the effect of retargeting advertisements is not proportional to the interaction between consumers and shopping malls, and has a nonlinear relationship. The results of this study can contribute to marketing methodologies and the online advertising literature.

*5.2. Theoretical Implications*

The theoretical implications of this study are as follows. First, this study is significant in that it presents various issues that arise when applying machine learning to marketing decision making in an online context and suggests effective methods. Prior research on machine learning for predictive purposes reports that marketing decision-making can be supported using various machine learning methodologies. However, research related to online consumer conversion prediction is relatively limited. In addition, there has not been enough research on the types of machine learning methodologies that are effective in the context of online consumer behavior prediction. This study expands our knowledge of online consumer behavior analysis methodology by comparing 8 machine learning model algorithms and 3 sampling methods in this context.

Second, the possibility of using machine learning in the marketing context was verified by using various XAI methodologies. In addition to the popularity of machine learning, various machine learning models are being applied in the context of marketing. However, as suggested in the work of Cui and Curry [1], machine learning models have difficulty interpreting the relationship between predictors and outcomes. Although the XAI methodology is being developed, there are not enough studies being applied in the context of marketing yet. This study connects the machine learning and marketing literature and expands marketing methodology by applying XAI from the perspective of global and individual consumers.

Third, by applying a machine learning model to the context of retargeting advertisements, we explored which consumers it is effective to target using retargeting advertisements. Retargeting advertisements are personalized advertisements and have a deep relationship with machine learning predictions, but sufficient research has not been presented thus far [3]. As a result of analyzing the machine learning model using XAI in this

study, surprisingly, variables representing interactions, such as page view and duration, which had a positive relationship with online conversion behavior in previous studies, did not show a linear relationship, but instead showed a nonlinear relationship. In addition, it was found that there is a difference in the effect of retargeting advertisements depending on the inflow channel. The exploratory results of this study offer new implications for retargeting advertising research.

### 5.3. Practical Implications

The practical implications of this study are as follows. First, practitioners can use the results of this study as a guide in applying machine learning to predicting online consumer behavior. One of the biggest challenges in applying machine learning methodologies to marketing work is improving accuracy. It takes a lot of time and money to compare different machine learning models and explore the parameters with the best performance. As a result of the analysis of this study, the ensemble model and over-sampling method were found to be the most effective methods. In addition, it is possible to reduce the time and effort required for model tuning by utilizing the Bayesian-parameter-tuning used in this study.

Second, using the XAI methodology presented in this study, the results of machine learning model analysis can be applied not only to channel budget allocation from a global perspective, but also to individual consumer channel management. The importance and impact of each inflow channel can be analyzed through the SHAP importance analysis, and furthermore, the appropriate investment amount for each channel can be calculated through the SHAP feature dependence analysis. In addition, from the perspective of individual consumers, by applying XGB Explainer, marketers will be able to take the most appropriate marketing action for heterogeneous individual consumers.

Third, this study can provide practical implications for the execution of retargeting advertisements. As a result of applying XAI in the context of retargeting advertising, we found that the effect of retargeting advertising is not proportional to the interaction between consumers and shopping malls, and has a nonlinear relationship. Therefore, marketers need to be careful to preferentially apply retargeting advertisements to consumers with the highest interaction indicators based on their intuition. This study provides implications that advertisement ROI can be increased by analyzing consumer characteristics appropriate for retargeting advertisements using a machine learning model.

### 5.4. Limitations of This Study and Future Research Directions

The limitations of this study and future research directions are as follows. First, this study utilized data from the Google Merchandise Store. This shopping mall sells Google's souvenirs, and there may be differences in consumer behavior from general online shopping malls. Also, in the case of a shopping mall that sells services rather than a product shopping mall, there may be a difference in the analysis result. Therefore, in future research, analysis using various online shopping mall data is needed. Second, in this study, eight major machine learning models were compared, and five candidate values for tuning parameters were set using the CARET package. However, the most suitable machine learning model may vary depending on models and tuning parameters not covered in this study. Therefore, in future studies, it is necessary to explore a wider range of parameter candidates together with machine learning models not covered in this study. Third, although the machine learning model was verified with OOB in this study, the identification problem exists because counterfactual dependence was not verified. Therefore, it is necessary to analyze the effects of machine learning models through field experiments in future studies. In addition, by utilizing causality machine learning, new implications can be provided for machine learning research in the marketing field. Finally, this study has a limitation in that it does not analyze marginal effects. This is because the interpretation of marginal effects constitutes the key aspect to evaluate the impact on the target: each unit increase in the independent variable increases/decreases the probability

of selecting the alternative "purchasing" by the value of the marginal effect expressed as a percentage. Therefore, it is necessary to provide economic interpretation of coefficients by analyzing marginal effects in future studies.

**Author Contributions:** Conceptualization, J.L.; methodology, J.L. and O.K.; writing—original draft preparation, J.L., O.J. and Y.L.; supervision, C.P. All authors have read and agreed to the published version of the manuscript.

**Funding:** This research received no external funding.

**Institutional Review Board Statement:** Not applicable.

**Informed Consent Statement:** Not applicable.

**Data Availability Statement:** Restrictions apply to the availability of these data. Data was obtained from Kaggle.com.

**Conflicts of Interest:** The authors declare no conflict of interest.

## Appendix A

**Table A1.** Bayesian-parameter-tuning results.

| Sampling | Round | Max. Depth | Min_Child | Subsample | Eta | Gamma | Colsample | Max_Delta | AUC | Nrounds |
|---|---|---|---|---|---|---|---|---|---|---|
| Original data | 1 | 3 | 5 | 0.67542 | 0.17485 | 0.10366 | 0.62363 | 2 | 0.86962 | 1000 |
| | 2 | 4 | 20 | 0.76645 | 0.01474 | 0.01829 | 0.43539 | 2 | 0.86840 | |
| | 3 | 5 | 23 | 0.68601 | 0.10140 | 0.01447 | 0.57288 | 3 | 0.86859 | |
| | 4 | 4 | 12 | 0.71403 | 0.07067 | 0.19889 | 0.72344 | 4 | 0.86946 | |
| | 5 | 3 | 3 | 0.57810 | 0.02189 | 0.06510 | 0.77619 | 2 | 0.86969 | |
| | 6 | 2 | 5 | 0.58433 | 0.22802 | 0.18413 | 0.51555 | 9 | 0.86765 | |
| Down-sampling | 1 | 2 | 30 | 0.58982 | 0.10479 | 0.0902 | 0.422182 | 2 | 0.859603 | 152 |
| | 2 | 2 | 9 | 0.595898 | 0.212168 | 0.117043 | 0.379914 | 9 | 0.859942 | |
| | 3 | 6 | 5 | 0.894569 | 0.048176 | 0.131024 | 0.383392 | 9 | 0.862621 | |
| | 4 | 3 | 2 | 0.769098 | 0.274687 | 0.2 | 0.373374 | 7 | 0.861747 | |
| | 5 | 3 | 27 | 0.869671 | 0.198249 | 0.04005 | 0.552015 | 10 | 0.86013 | |
| | 6 | 6 | 6 | 0.520637 | 0.041795 | 0.191857 | 0.44203 | 7 | 0.861421 | |
| Over-sampling | 1 | 3 | 23 | 0.631274 | 0.079454 | 0.050344 | 0.749949 | 3 | 0.993592 | 667 |
| | 2 | 3 | 5 | 0.891644 | 0.12731 | 0.189899 | 0.792452 | 5 | 0.993699 | |
| | 3 | 5 | 20 | 0.681852 | 0.242676 | 0.096076 | 0.365477 | 6 | 0.993591 | |
| | 4 | 4 | 3 | 0.80138 | 0.066665 | 0.090604 | 0.651716 | 6 | 0.993813 | |
| | 5 | 5 | 6 | 0.556146 | 0.046641 | 0.002572 | 0.514211 | 3 | 0.993759 | |
| | 6 | 5 | 1 | 0.638261 | 0.209523 | 0.108228 | 0.684999 | 2 | 0.993699 | |

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
