# Peer review of "A Comparison and Interpretation of Machine Learning Algorithm for the Prediction of Online Purchase Conversion"

_jtaer, doi:10.3390/jtaer16050083_

Round 1

Reviewer 1 Report

The paper "A Study on Online Purchase Conversion Prediction Using Machine Learning" has several problems, namely:

  • The study does not present the mathematical formalization of the different models used;
  • Data is private; thus, other researchers cannot test their own deep learning artificial neural networks on your data (this is a paradox because you submit a paper to an open access journal, but you keep the data private);
  • The study does not provide nothing new (i.e., machine learning models used in the classification analysis are basic rather than being new and personalized);
  • The data suffer from the identification problem (i.e., despite authors apply regularization and other techniques, they need to confirm causality by analyzing treatment effects through a differences-in-differences approach);
  • The  economic interpretation of coefficients is incorrect (i.e., you have a R package that gives you coefficients, but you do not know how their interpretation is made; you really need to read Daniel McFadden - i.e., 'the father of discrete choice analysis' (https://eml.berkeley.edu/~mcfadden/iatbr00.html) - because you need to compute not only coefficients but also average marginal effects (and interpret both).

Overall, the paper seems to be developed by Masters/PhD students. This is not necessarily a bad thing, but the journal is indexed in SSCI and focused on the field of Business (instead of being focused on Computer Science).

Therefore, the professionals in question need to provide a valid econometric (discrete choice) analysis because they were unable to provide a state-of-the-art contribution from a technical point of view (i.e., in other words, I believe that, if you decide to submit this paper to a machine learning journal, it will be rejected).

With that being said, the paper is still not ready to be published. Do not be discouraged by the present report. I wish you good luck and hope that you have the patience to improve the "economic" part of the paper given that the "technical (or computer science)" part of the paper is insufficient to ensure a publication.

In summary, if the editor-in-charge decides that your paper should go to a 'revise and resubmit' stage, you need to correctly perform section 4.3 (Research Question 3 in its current form is incorrect) and clearly declare that it is not the goal of the paper to improve the technical content related to machine learning, but to strictly focus on the economic issue of  interpreting online consumer behavior classification results (your paper is not a predictive exercise, but rather a classification task). 

Yours sincerely.

Author Response

Thank you very much for your constructive feedback on our manuscript. We also appreciate the issues your raised and, in our revised manuscript, we did our best to address them and incorporate your suggestions. The way we have dealt with the various concerns you raised is outlined in detail below.

Comment 1: The study does not present the mathematical formalization of the different models used;
Response: Thank you for pointing this out. Since the machine learning model covered in Research Question 1 is already a popular model, we added a reference without presenting mathematical formalization. Since the XGB model is used in Research Questions 2 and 3 and is also related to XGB Importance analysis, the objective function and the Gain function are presented. In addition, the algorithm of the CARET package, the methodology of research question 1, was presented (p. 10-11).

Comment 2: Data is private; thus, other researchers cannot test their own deep learning artificial neural networks on your data (this is a paradox because you submit a paper to an open access journal, but you keep the data private);
Response: Thank you for pointing this out. We agree with this comment. However, the data we use can be accessed through Kaggle.com (https://www.kaggle.com/c/ga-customer-revenue-prediction/data). In addition, although it was 2019, before open access, the journal published a paper with the same data (Kakalejcík et al., 2019). Also, we would like you to understand that online consumer data is very difficult to secure as open data. Most of the research is conducted on private data because it contains behavioral data of consumers with privacy issues.

Comment 3: The data suffer from the identification problem (i.e., despite authors apply regularization and other techniques, they need to confirm causality by analyzing treatment effects through a differences-in-differences approach);

Response: Thank you for pointing this out. Because typical machine learning models learn data based on correlations, it is difficult to analyze causal relationships. In addition, the scope of this study is to explore and review the applicability of machine learning settings for online consumer behavioral data. Therefore, we believe that it is outside the scope of this study to verify new hypotheses using the differences-in-differences approach. However, as you have pointed out, the machine learning verification we used cannot be free from the identification problem, so we have presented the content in its limitations and future research directions (p. 18).

Comment 4: The economic interpretation of coefficients is incorrect (i.e., you have a R package that gives you coefficients, but you do not know how their interpretation is made; you really need to read Daniel McFadden - i.e., 'the father of discrete choice analysis' (https://eml.berkeley.edu/~mcfadden/iatbr00.html) - because you need to compute not only coefficients but also average marginal effects (and interpret both).

Response: Thank you for pointing this out. We have some agreement with you. According to your point of view, the newly written section 4.3 presents the SHAP and log odds functions. However, the context of this study is a machine learning model, and the machine learning model is considered a black box model. Explainable Artificial Intelligence (XAI) methodology has emerged to explore these black boxes, and we think that it is difficult to apply the criteria of the traditional econometric model. According to the study of Tremblay et al. (2021), there is some difference between traditional model evaluation and machine learning model evaluation.

Comment 5: In summary, if the editor-in-charge decides that your paper should go to a 'revise and resubmit' stage, you need to correctly perform section 4.3 (Research Question 3 in its current form is incorrect) and clearly declare that it is not the goal of the paper to improve the technical content related to machine learning, but to strictly focus on the economic issue of  interpreting online consumer behavior classification results (your paper is not a predictive exercise, but rather a classification task).
The study does not provide nothing new (i.e., machine learning models used in the classification analysis are basic rather than being new and personalized);

Response: Thank you for pointing this out. We agree with this comment. As you suggested, we have strengthened the interpretation of online consumer behavior. To this end, the content of Section 4.3 has been completely rewritten. Specifically, factors influencing retargeting ad conversions that were not sufficiently covered in previous studies were explored through machine learning model analysis. For the XAI, SHAP analysis was also added. SHAP analysis allowed us to analyze the impact and direction of individual variables on the likelihood of conversion in a machine learning model. Based on the analysis results, we analyzed the factors that influence the retargeting conversion and found a nonlinear relationship. And based on these results, the theoretical and practical implications were reinforced (p.13-15).

References

Kakalejcík, L., Bucko, J.; Vejacka, M. Differences in Buyer Journey between High- and Low-Value Customers of E-Commerce Business. J. Theor. Appl. El. Comm. 2019, 14, 47-58.

Tremblay, M. C., Kohli, R.; Forsgren, N. THEORIES IN FLUX: REIMAGINING THEORY BUILDING IN THE AGE OF MACHINE LEARNING, MIS Q. 2021, 45, 455-459.

Reviewer 2 Report

The introduction is not very useful. Therefore, the introduction should be extended very carefully. The introduction section should be rewritten again. The introduction should highlight the study's novelty and motivation and put some literature without any useful explanation; in fact, the introduction should be clearly stated research questions and targets first. Then answer several questions: Why is the topic important (or why do you study on it)? What are the research questions? What has been studied? What are your contributions? Why is it to propose this particular method? This study's major defect is the debate or argument is not clearly stated in the introduction session.

I would suggest the author improve your theoretical discussion and arrives at your debate or argument. In addition, the background introduction should be condensed. The literature review is not presented in a good structure, and at the end of LR, you should come out with a paragraph to conclude your discussion, in this paragraph, you can highlight the novelty of your study also, it means what the LR has done and what you want to do. The literature review must highlight the novelty and contribution of the study, but these sections, which the authors provided only are related works and not literature review. Authors must carefully revise these sections.

There are several grammatical errors in the paper. Expert opinion details are not provided.

Author Response

Response to Reviewer 2

Thank you very much for your constructive feedback on our manuscript. We also appreciate the issues your raised and, in our revised manuscript, we did our best to address them and incorporate your suggestions. The way we have dealt with the various concerns you raised is outlined in detail below.

Comment 1: The introduction is not very useful. Therefore, the introduction should be extended very carefully. The introduction section should be rewritten again. The introduction should highlight the study's novelty and motivation and put some literature without any useful explanation; in fact, the introduction should be clearly stated research questions and targets first. Then answer several questions: Why is the topic important (or why do you study on it)? What are the research questions? What has been studied? What are your contributions? Why is it to propose this particular method? This study's major defect is the debate or argument is not clearly stated in the introduction session.

Response: Thank you for pointing this out. We agree with this comment. The introduction has been supplemented. Specifically, we presented the literature we intend to contribute and suggested related studies. In addition, what we found about the proposed gap and supplemented the theoretical contributions (p. 3).

Comment 2: I would suggest the author improve your theoretical discussion and arrives at your debate or argument. In addition, the background introduction should be condensed. The literature review is not presented in a good structure, and at the end of LR, you should come out with a paragraph to conclude your discussion, in this paragraph, you can highlight the novelty of your study also, it means what the LR has done and what you want to do. The literature review must highlight the novelty and contribution of the study, but these sections, which the authors provided only are related works and not literature review. Authors must carefully revise these sections.

Response: Thank you for pointing this out. We agree with this comment. Following the proposal, a paragraph explaining the limitations of previous research and the contribution of this research has been added at the end of LR (p. 5-6). In addition, a comparison table between related literature and this study has been added to highlight these contributions (Table 1).

Comment 3: I would suggest the author improve your theoretical discussion and arrives at your debate or argument. In addition, the background introduction should be condensed. The literature review is not presented in a good structure, and at the end of LR, you should come out with a paragraph to conclude your discussion, in this paragraph, you can highlight the novelty of your study also, it means what the LR has done and what you want to do. The literature review must highlight the novelty and contribution of the study, but these sections, which the authors provided only are related works and not literature review. Authors must carefully revise these sections.

Response: Thank you for pointing this out. We agree with this comment. Following the proposal, a paragraph explaining the limitations of previous research and the contribution of this research has been added at the end of LR (p. 5-6). In addition, a comparison table between related literature and this study has been added to highlight these contributions (Table 1).

Comment 3: There are several grammatical errors in the paper. Expert opinion details are not provided.

Response: Thank you for pointing this out. We agree with this comment. The manuscript's errors were reviewed again. As the manuscript gets closer to publication, we are considering professional proofreading services.

Reviewer 3 Report

This study aims to identify a suitable machine learning model for predicting online consumer behavior and a good data sampling method for predicting online consumer behavior. The results show the performance of the ensemble model, the XGBoost model, was the best for predicting the purchase conversion of online consumers. The topic of this article is of interest to wide readers. The paper's argument built on an appropriate base of theory and methods employed in the paper are appropriate, however, the list of references needs an update by including more recent articles.

Author Response

Response to Reviewer 3

Thank you very much for your constructive feedback on our manuscript. We also appreciate the issues your raised and, in our revised manuscript, we did our best to address them and incorporate your suggestions. The way we have dealt with the various concerns you raised is outlined in detail below.

Comment 1: This study aims to identify a suitable machine learning model for predicting online consumer behavior and a good data sampling method for predicting online consumer behavior. The results show the performance of the ensemble model, the XGBoost model, was the best for predicting the purchase conversion of online consumers. The topic of this article is of interest to wide readers. The paper's argument built on an appropriate base of theory and methods employed in the paper are appropriate, however, the list of references needs an update by including more recent articles.

Response: Thank you for pointing this out. We agree with this comment. We have reviewed the literature and added the latest research. In addition, related research tables have been added to the literature review part (p. 5-6).

For example.

  1. Li, H.; Kannan, P. K. Attributing conversions in a multichannel online marketing environment: An empirical model and a field experiment, Marketing Res. 2014, 51, 40-56.
  2. Xu, L., Duan, J. A.; Whinston, A. Path to purchase: A mutually exciting point process model for online advertising and conversion, Sci. 2014, 60, 1392-1412.
  3. Kireyev, P., Pauwels, K.; Gupta, S. Do display ads influence search? Attribution and dynamics in online advertising, J. Res. Mark. 2016, 33, 475-490.
  4. Abhishek, V., Hosanagar, K.; Fader, P. S. Aggregation bias in sponsored search data: The curse and the cure, Sci. 2015, 34, 59-77.
  5. Meng, Y., Yang, N., Qian, Z.; Zhang, G. What Makes an Online Review More Helpful: An Interpretation Framework Using XGBoost and SHAP Values, theor. appl. electron. commer. res. 2021, 16, 466-490.

Reviewer 4 Report

This study derives the following research questions through literature reviews on online  consumer conversion and machine learning. 1) What is the suitable machine learning model for  predicting online consumer behavior? 2) What is the good data sampling method for predicting  online consumer behavior? 3) Can we interpret machine learning's online consumer behavior prediction results? The authors aim to contribute at comparing the performance of eight machine learning model algorithms and exploring interpretation methods for prediction results.

In general, the paper is well structured and written.

The abstract is clear, it presents the object of research and the  results.

The authors prove to know well the extant literature.

The methodology seems sound.

The results and interpretations are correct but should refer to the results of previous studies.

The conclusions should be further developed in order to explore the main advantages of the approach herein followed. What are the policy implications? What are the main differences regarding the outcomes obtained with this approach and the other approaches available in the scientific literature?

Author Response

Response to Reviewer 4

Thank you very much for your constructive feedback on our manuscript. We also appreciate the issues your raised and, in our revised manuscript, we did our best to address them and incorporate your suggestions. The way we have dealt with the various concerns you raised is outlined in detail below.

Comment 1: This study derives the following research questions through literature reviews on online consumer conversion and machine learning. 1) What is the suitable machine learning model for predicting online consumer behavior? 2) What is the good data sampling method for predicting online consumer behavior? 3) Can we interpret machine learning's online consumer behavior prediction results? The authors aim to contribute at comparing the performance of eight machine learning model algorithms and exploring interpretation methods for prediction results.

In general, the paper is well structured and written.

The abstract is clear, it presents the object of research and the results.

The authors prove to know well the extant literature.

The methodology seems sound.

The results and interpretations are correct but should refer to the results of previous studies.

The conclusions should be further developed in order to explore the main advantages of the approach herein followed. What are the policy implications? What are the main differences regarding the outcomes obtained with this approach and the other approaches available in the scientific literature?

Response: Thank you for pointing this out. We agree with this comment. The research implication’s part was rewritten. Specifically, it has been prepared in detail by dividing it into theoretical and practical implications (p. 17-18). In addition, many parts have been supplemented in the interpretation of the results.

Round 2

Author Response

Thank you very much for your constructive feedback on our manuscript. We also appreciate the issues your raised and, in our revised manuscript, we did our best to address them and incorporate your suggestions. The way we have dealt with the various concerns you raised is outlined in detail below.

Comment 1: In this type of models, first, you need to know what your base outcome is in order to make an interpretation of coefficients. In your case study, the dependent variable y is a categorical and unordered variable, so that a given individual selects only one alternative. Categories are called alternatives and are coded as j = 0,1 because they take one out of two possible outcomes. That is, the target or dependent variable is a binary variable, which means that the interpretation respects a binary discrete choice modelling approach (1 if purchase occurs; 0 if purchase does not occur). The previous ‘bold’ part must be written in the main text of your study.

Response: Thank you for pointing this out. We have added the sentence you suggested (p. 9). Thank you for explaining the direction of supplementation in detail.

Comment 2: Now, the last and most important part that is missing in your study is that you must interpret coefficients and compute marginal effects. Interpreting coefficients is different from making a variable importance analysis. For instance, the coefficient associated with duration = 705 in Fig. 6 is equal to -0.28. Assuming that 705 is a measure whose unit is in seconds, this means that, in comparison to the base alternative (that is, in comparison to the decision of not purchasing), an increase in the independent variable (that is, an increase in the duration of the purchase from 705 seconds to 706 seconds) makes the individual’s decision of purchasing less likely. Hence, the interpretation of this coefficient suggests that extra time spent by an individual on purchasing decreases the likelihood of making a purchase. This is the kind of interpretation that is missing in your study and that is showing to other specialists that you only have a strong background in computer science. However, JTAER is a Business journal! Moreover, as I already told in the first review, this study does not improve the state-of-the-art of the machine learning field because I also have a strong background in machine and deep learning, which means that I can easily distinguish between good and redundant academic works. Tremblay et al. (2021) claim that there is some difference between traditional model evaluation and machine learning model evaluation, but they never say in any part of their paper that researchers are unable to perform an interpretation of coefficients. Hence, please, do not use an academic study to excuse yourself from not interpreting coefficients. Your study has the potential of becoming a reference in JTAER, which means that you should raise the bar as much as possible rather than just publishing another paper. Hence, I only believe that this paper is worth publication if and only if you improve the economic interpretation of the model. My suggestion is that you should copy and paste my previous interpretation of the coefficients related to “duration” (if the unit of measure is seconds) or you can simply do the interpretation of a dummy variable acknowledging my suggestion in a footnote. I am here to help your study becoming a reference, not to block your publication.

Response: Thank you for pointing this out. We have added the sentence you suggested (p. 16). Thank you for explaining the direction of supplementation in detail. Also, thank you for introducing Daniel McFadden to me. I'm reading hard these days.

Comment 3: Finally, it is also important to tell you that the interpretation of coefficients, by itself, is irrelevant. For economists, the relevant part is the interpretation of marginal effects, which you did not compute in the study. This is because the interpretation of marginal effects constitutes the key aspect to evaluate the impact on the target: each unit increase in the independent variable increases/decreases the probability of selecting the alternative “purchasing” by the value of the marginal effect expressed as a percentage. My suggestion here is to write in the conclusions that a limitation of the study is that you do not analyze marginal effects, which is a task left for the future.

Response: Thank you for pointing this out. We have added the sentence you suggested (p. 18). Thank you for explaining the direction of supplementation in detail.

Thank you very much again for your thoughtful and highly constructive review.

Reviewer 2 Report

Authors have responded to my comments adequately on the earlier draft. I have no further comments. 

Author Response

Comment: Authors have responded to my comments adequately on the earlier draft. I have no further comments. 

Response: Thank you very much for your constructive feedback on our manuscript.  I checked English grammar and spelling.

Thanks again.